# Detection of mpox and other orthopoxviruses using a lateral flow device as a point-of-care diagnostic

Stephen M. Laidlaw,[1,2] David Ulaeto,[3] Steve Lonsdale,[3] Graeme Clark,[3] Rebecca Sumner,[4] Thomas Edwards,[5] Emily Adams,[5,6] Anne-Sophie Logist,[7] Bram Van Holm,[7] Carlos Maluquer de Motes,[4] Peter Horby,[1] Piet Maes,[7] Miles W. Carroll[1,2]

**ABSTRACT** In 2022, the World Health Organization declared the worldwide outbreak of mpox to be a public health emergency of international concern. The causative monkeypox virus (MPXV) belonged to clade IIb and is transmitted through sexual contact with a low case fatality rate (0.1%), which, together with under-detection, all contributed to a rapid global spread particularly within the MSM (men who have sex with men) community. As MPXV clade II remains circulating worldwide, a new outbreak of the more fatal clade I disease has been declared in Central and East Africa, and remains uncontrolled in part due to the lack of point-of-care (POC) diagnostics for rapid decisions on treatment and self-isolation. To address the lack of POC solutions for mpox, we have designed and evaluated an orthopoxvirus-specific lateral flow device (LFD) that could be used for the diagnosis of mpox. Using an LFD comprising four monoclonal antibodies against the A27 protein, we demonstrate sensitivity to $3 \times 10^5$ pfu/mL. This sensitivity is expected to be sufficient for the detection of MPXV from lesion sites and may also be sufficient for other sample types such as saliva and urine. We found that the presence of guanidinium thiocyanate, a common ingredient in inactivating viral transport media, masked the LFD antigen, resulting in false negatives. POC diagnosis of mpox may be possible using an LFD to reduce delays arising from sample shipment to centralized laboratory testing facilities. In order to achieve this, our work demonstrates that an LFD-optimized buffer is required, as the sample collection buffer may have a detrimental impact on sensitivity for clinical material.

**IMPORTANCE** Mpox cases have dramatically increased both in traditionally monkeypox virus endemic countries and also worldwide. This increase comes at a time when immunity derived from smallpox vaccination is no longer available. Diagnosis of mpox is complicated due to both disease presentation and the availability of local diagnostic laboratories. The availability of a point-of-care diagnostic tool such as an lateral flow device (LFD) would play an important role to both diagnose and prevent onward transmission. This manuscript provides developers and assessors with key data for defining true sensitivity and specificity of a successful LFD in addition to buffer conditions for sample collection.

**KEYWORDS** mpox, LFD, diagnostics

I n May 2022, a worldwide outbreak of mpox caused widespread infections in multiple countries outside the Central and West African endemic range of the causative agent monkeypox virus (MPXV) (1). The World Health Organization (WHO) declared the outbreak a public health emergency of international concern (PHEIC) in July 2022 (2). By the time WHO declared the public health emergency over in May 2023, more than 87,000 cases and 140 confirmed deaths had been reported across 111 countries. As of June 2024, these numbers had risen to 99,176 confirmed cases and 208 deaths in

Address correspondence to Stephen M. Laidlaw, stephen.laidlaw@well.ox.ac.uk, or Miles W. Carroll, miles.carroll@ndm.ox.ac.uk.

Stephen M. Laidlaw and David Ulaeto contributed equally to this article. Author order was determined alphabetically.

Piet Maes and Miles W. Carroll are joint senior authors.

The authors declare no conflict of interest.

See the funding table on p. 12.

116 countries. Fifteen months later, the WHO has declared another PHEIC (3) following the rapid spread of mpox across Central Africa. This escalation was a consequence of the Africa CDC declaring a public health emergency on 13 August 2024 (4). In these regions, particularly in the Democratic Republic of the Congo (DRC), clade I MPXV is currently on the increase (5). This clade has been reported to affect young children and has also been associated with sexual contact (6). There are unconfirmed cases in West Africa, and additionally, a novel clade I sub-lineage (clade Ib) has recently emerged in the DRC, primarily in young sex workers (7), and has been reported to be associated with sexual contact in both men who have sex with men and heterosexual networks. New cases of mpox continue to be diagnosed outside Africa, with some reports suggesting possible reinfections (8, 9). The global outbreak was initially caused by clade IIb MPXV and appeared to be facilitated by a novel transmission mode involving intimate or sexual contact (10). Currently, there are confirmed extended human transmission chains involving a similar route with the more virulent clade I viruses (historical case fatality rates up to 11%) (11). This suggests that the novel mode of human-to-human transmission may be applicable to both clades of MPXV. Coupled with the continued occurrence of clade II infections outside the endemic regions, it appears that mpox is establishing as a non-zoonotic human infection.

As an *Orthopoxvirus* (OPXV), MPXV shares significant genetic and antigenic characteristics with other old-world OPXVs such as vaccinia virus (VACV) and the causative agent of smallpox, variola virus. This similarity underlies the success of smallpox vaccination with VACV (12). However, the traditional smallpox vaccine is no longer considered suitable for widespread use due to a high incidence of adverse events including life-threatening or fatal complications in people with immunodeficiencies. Safer, non-replicating VACV-based smallpox vaccines have been developed (13–18) and some have been licensed for mpox across multiple jurisdictions (19). In the UK, people at high risk of mpox are recommended to be vaccinated, "off-label," with the JYNNEOS (also named IMVANEX in Europe and IMVAMUNE in the USA) modified vaccinia Ankara-Bavarian Nordic (MVA-BN) vaccine, which is based on the highly attenuated modified vaccinia virus Ankara (MVA) strain (20, 21).

A report to the G7 group of industrialized nations in June 2021 outlined the "100 Days Mission" for the world to respond to Disease X by producing medical countermeasures, including diagnostics within 100 days (22). Robust, rapid diagnostic tests are critical for preventing spread of pandemic diseases, and the Coalition for Epidemic Preparedness Innovations has identified this as a key area for research to help meet the G7 Disease X mission (23). It is therefore important to prioritize the development or repurposing of current technologies for point-of-care (POC) diagnostics for an emerging disease such as mpox.

PCR testing is currently the gold standard for viral infectious disease diagnostics. For mpox, this typically involves using swabs or samples from affected areas or lesions, followed by laboratory-based downstream processing and analysis. However, this method is not suitable for home or lower-resource POC diagnosis, and the resulting delays can hamper efforts to slow or halt disease spread. This is especially relevant in lower-income countries where there is a clear need for inexpensive, low-infrastructure POC diagnostics that can provide reliable diagnosis within a short timeframe. Current CDC guidelines recommend that testing for mpox is only carried out for people with a rash consistent with mpox (24, 25). In the current outbreak, mpox lesions are reported to often be atypical (26, 27). This may be due to undiagnosed cases of co-infection (24, 28, 29). The presence of a co-incident rash that may present at the same location as the mpox rash can result in misdiagnosis or under-reporting following PCR, due to either lower sensitivity or misidentification.

Global experience during the coronavirus disease 2019 (COVID-19) pandemic demonstrated the utility of antibody-based lateral flow devices (LFDs) for rapid, routine diagnosis without the need for trained personnel or laboratory facilities. COVID-19 LFDs were instrumental in reducing the transmission of severe acute respiratory syndrome

coronavirus 2 (SARS-CoV-2) as people interacted to perform essential activities during the pandemic. For instance, the Innova LFD, procured by the UK government, was reported to have a limit of detection (LOD) of 390 plaque-forming units (pfu) per milliliter when using saliva samples spiked with SARS-CoV-2, which is equivalent to a cycle threshold (Ct) value of 25.2 (30). At these antigen levels (Ct <25), LFDs had a sensitivity greater than 90%, suggesting that an LFD approach could be useful for preliminary diagnosis of mpox.

According to the FIND database, there are currently 12 lateral flow tests for mpox under development or with "regulatory achievement" classified as true POC (31), although no mpox LFDs have been FDA-approved or received emergency use authorization (according to Devices@FDA). Prior to the global outbreak, the only readily available commercial LFD that targets OPXVs was the Tetracore antigen detection assay (Tetracore, Rockville, MD). This LFD has an LOD of $10^7$–$10^8$ viral particles per milliliter, using a lateral flow assay (LFA) reader (Zachary Polhemus, Tetracore, personal communication). Assuming 100 particles per pfu, the LOD equates to $10^5$–$10^6$ pfu/mL or $1.5 \times 10^4$–$1.5 \times 10^5$ pfu/LFA. Townsend et al. (32) have demonstrated LOD for VACV and MPXV of $10^6$ and $1.5 \times 10^5$ pfu, respectively, using this system. However, this assay is not suitable for POC use because sample processing required sonication for swab material and dry ice/ethanol bath freezing followed by pestle grinding and sonication for scab material.

OPXV replication takes place within the infected cell cytoplasm producing two distinct forms, so-called intracellular mature virus (IMV) and enveloped virus (EV). EVs are extracellular, are wrapped with a double membrane, and constitute 5%–20% of the virion progeny (33), whereas the remaining progenies are wrapped with a single membrane and remain intracellular to become an IMV (34–36). Both mature forms have unique surface antigens, and to maximize utility of an LFD, it is assumed that antigens found on both forms should be targeted.

There are 22 membrane proteins found on IMV particles (37), of which the majority are also found beneath the envelope of EV. Of these, the most abundant is A27, ranked fifth by virion mass (4.09%). The A27 protein sequence is highly conserved, with 94% amino acid identity between VACV and MPXV. The highly conserved amino acid sequence of A27, its localization on the outside of the IMV and its abundance, all combine to make it an ideal antigen for LFD design.

The sensitivity of OPXV LFDs is established using cultured and, in some cases, purified virus, allowing reproducible correlation between quantity of accessible antigen (LFD signal), quantity of infectious virus, and PCR Ct value. However, in clinical samples, these correlations are uncertain because virus nucleic acid can persist in the absence of detectable infectious virus; and virus antigen may be present in excess of infectious virions and/or persist in the absence of detectable infectious virions. Using PCR, mpox DNA can be detected at lesion sites and is also found in saliva, semen, and other clinical samples (38, 39). Using dermal lesion material and oropharyngeal swabs, a correlation between viral DNA quantity (Ct) and virus infectivity in cell culture has shown that Ct values ≥35 (corresponding to viral DNA ≤4,300 copies/mL) predict no or very low infectivity (38). Using the same samples, a ratio of 1 pfu to 172 DNA copies was calculated (38).

We have previously demonstrated a prototype LFD based on the VACV A27 protein with an LOD between $3 \times 10^4$ and $1 \times 10^5$ pfu using purified MVA strain VACV, with a 1 log reduction in sensitivity in saliva (40). The highest concentrations of MPXV DNA are found in skin lesions, and this is assumed to also contain the highest concentration of virus and virus antigen. In this study, we demonstrate that antibodies used in the LFD recognize all OPXVs tested, and do not recognize viruses from different poxvirus genera. We furthermore demonstrate equivalent sensitivity of the LFD for clade IIb MPXV and VACV strain MVA using purified virus. The LOD is predicted to be sufficient to detect MPXV in clinical samples, assuming the correlation between Ct value and infectious virus titer in clinical samples provided by Paran et al. (38). We have also demonstrated that the

LFD can be used to detect MPXV in clinical samples, provided the samples are stored in LFD-compatible transport medium.

## MATERIALS AND METHODS

### Virus strains

VACV strain IHD-J (VR-156), rabbit (Shope) fibroma virus (RFV) strain OA (VR112), fowlpox virus (FWPV) strain FH (VR-229), pseudocowpox virus (PCPV) strain TJS (VR-634), myxoma virus (MYXV) strain Lausanne (VR-115), swinepox virus (SWPV) strain Kasza (VR-363), and Tanapox virus (TANV) strain Davis (VR-937) were purchased from the American Type Culture Collection. Camelpox virus (CMLV) strain Somalia, cowpox virus (CPXV) strain Brighton Red and MPXV strain Zaire-79 were the kind gifts of Dr. J. Huggins. VACV-MVA was supplied by The Jenner Institute Viral Vector Core facility, University of Oxford. MPXV clade IIb lineage B (MPXV_CVR_S1, mpox2022) and lineage A (MPXV_UK_P3, mpox2018) were sourced from the University of Glasgow Centre for Virus Research and the UK Health Security Agency (UKHSA), respectively. Viruses were cultured in standard cell lines BSC40 (VACV, MPXV, CPXV, CMLV, TANV), RK-13 (RFV, MYXV), MDBK (PCPV), SK (SWPV), DF1 (VACV-MVA), or chick embryo fibroblasts (FWPV).

### Monoclonal antibodies (mAbs)

mAbs were raised using standard protocols after immunization of BALB/c mice. Briefly, CsCl gradient-purified IMV virions of VACV strain IHD-J were disrupted in a reducing buffer for SDS-polyacrylamide gel electrophoresis (SDS-PAGE). After SDS-PAGE, the 14 Kd protein band representing A27 was excised from a Coomassie-stained gel, washed in $H_2O$, and ground with a disposable pestle in a microcentrifuge tube. Ground material was emulsified in complete Freund's adjuvant and used for immunizations, with booster immunization using the same antigen emulsified in incomplete Freund's adjuvant. Hybridoma culture supernatants were assessed by enzyme-linked immunosorbent assay (ELISA) using ultraviolet (uv)-inactivated CsCl gradient-purified VACV IMV.

### ELISAs

Horseradish peroxidase-based ELISAs used either uv-inactivated or gamma-ray-inactivated viruses, except for ACDP1 viruses which were not inactivated. Where uv or gamma-ray inactivation was routinely employed, reactivity against viable virus was confirmed under appropriate containment. All ELISAs were in capture format using a ployclonal rabbit anti-VACV IMV antibody as capture reagent.

### LFD assay

VACV-MVA-infected cell lysates or purified, titered VACV-MVA or MPXV were serially diluted in assay buffer, and 100 µL was added to the LFD. The LFD results were recorded at 20 minutes after application of sample.

### MVA cell infection assay

DF1 cells were seeded at $5 \times 10^5$ cells per well in a six-well plate. Following 24 hours, the cells were infected with VACV-MVA at a multiplicity of infection (MOI) of 1. Following 24 hours, incubation media was removed, the cells were washed with phosphate-buffered saline (PBS), and 1 mL lysis buffer was added. The six-well plate was incubated for 5 minutes and lysis was monitored using a microscope. The cells were removed using a cell scraper, and the lysed cells vortexed and then centrifuged to remove cellular debris in preparation for dilution and application to LFDs.

### Real-time PCR assay

Lesion swabs were taken from MPXV-infected patients attending University Hospitals Leuven (UZ Leuven), Belgium, and stored in inactivating viral transport medium

(unknown composition). All patients were confirmed to be infected with MPXV clade IIb B.1 by reverse transcription-polymerase chain reaction (RT-PCR).

Following Ct determination by RT-PCR, the samples were stratified by Ct into low, medium, and high Ct and a subset (100 samples; 40 low Ct [between 10 and 20], 20 medium Ct [between 20 and 30], 20 high Ct [between 30 and 40], and 20 negatives [chickenpox or HPV positive]) was used to calculate the sample genome equivalence.

Assays were carried out following the method developed by Sklenovska et. al. (41).

## RESULTS

### Antibody characterization

Four anti-A27 mAbs were selected following screening in ELISA-based assays. Titration against VACV and MPXV (Zaire) showed a dose response effect with responses approximately half a log higher against VACV than against MPXV (Fig. 1A). These mAbs were further characterized against an OPXV panel comprising CMLV, CPXV, MPXV, and VACV. All four mAbs gave comparable absorbance readings except mAb Rub.5.17.35 which consistently failed to bind CMLV. In order to ensure that the mAbs only recognized orthopoxviruses, they were screened against a panel of other poxviruses from other poxvirus genera: RFV (*Leporipoxvirus*), FWPV (*Avipoxvirus*), PCPV (Milker's nodule virus, *Parapoxvirus*), MYXV (*Leporipoxvirus*), SWPV (*Suipoxvirus*), and TANV (*Yatapoxvirus*). All of the non-OPXVs screened gave absorbance readings similar to the negative control.

### Limit of detection

The LOD of a bespoke LFD incorporating the four anti-A27 mAbs was determined using gradient-purified VACV strain MVA in a titration of half-log serial dilutions in clinical sample buffer (CSB) across virus concentrations ranging from $10^7$ to $10^{3.5}$ pfu per sample. The control band (C) was consistent for all samples and the presence of viral antigen is

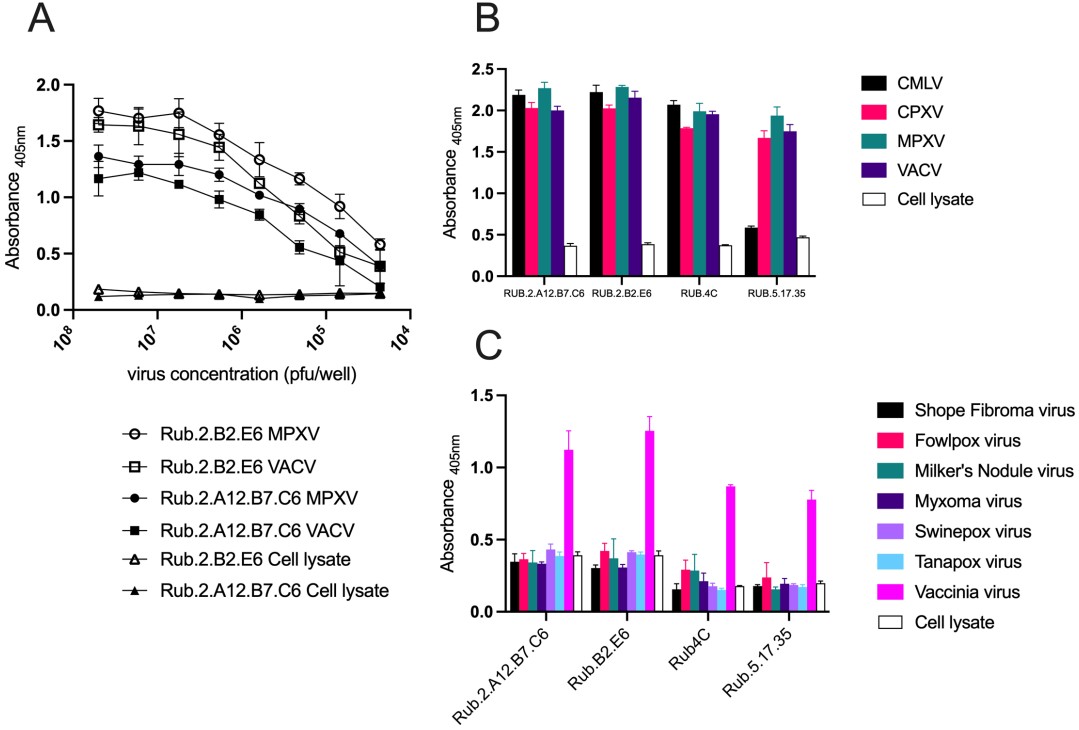

**FIG 1** Characterization of A27 monoclonal antibodies. (A) Titration of two mAbs; Rub.2A12.B7.C6 (filled symbols) and Rub.2.B2.E6 (open symbols) against vaccinia virus (circles) and mpox virus (squares). ELISA plates were coated with cell culture supernatant containing either titrated VACV, MPXV, or control supernatant. (B) Capture ELISA showing mAb binding to orthopoxviruses (CMLV, CPXV, MPXV, and VACV). (C) Capture ELISA showing mAb binding to other poxviruses (SFV, FWPV, Milkers nodule virus, MYXV, SWPV, TANV, and VACV).

clearly detected at the test position on the LFD between $10^7$ to $10^{4.5}$ pfu per sample (Fig. 2). This gives an LOD of $3 \times 10^4$ pfu which is consistent with our previous results (40). We defined this LOD as the lowest amount of virus that resulted in a visible band when viewed by eye. Confirmation of the LOD was carried out in all subsequent experiments ($n$ = 4) as an internal control (data not shown).

To improve the LOD of the test, virus titration was repeated with a range of buffers, formulated with different detergents (Table 1), supplied by the LFD manufacturer (BBI). All buffers gave a distinguishable band at $10^{4.5}$ pfu, but no bands were visible at $10^4$ pfu. Thus, these buffers did not improve the sensitivity of the LFD (Fig. 3). At a concentration of $10^5$ pfu per sample, a clear distinction was seen between the buffers. Those buffers containing higher concentrations of detergents (either Triton X-100, SDS, or Tween 80) produced darker test bands, and in the case of buffer 1 (containing 0.5% Triton X-100) produced a sharper test band.

## Detection of cell-associated MVA

Most OPXV transmit through the skin and the virus is found in infected skin rashes and mature pustules and pocks. To assess the performance of the LFD to detect virus in cellular milieu, we infected chicken fibroblast DF1 cells with MVA virus for 24 hours, after which cell monolayers were washed, and the cells lysed with buffers 1–7 before application to LFDs. All of the conditions tested detected virus from the infected cells (Fig. 4). As with the previous experiment, buffer 1 gave the strongest band. The substitution of a freeze thaw step in PBS for lysis buffer did not improve antigen detection (sample 9). Interestingly, the presence of SDS in samples 2, 3, 4, and 5(*) inhibited the detection of antigen, relative to other buffers, to a far greater extent than was observed with gradient-purified virus.

## Detection of MPXV

To test the sensitivity of the LFD on MPXV, sucrose cushion-purified preparations of two clade IIb MPXV isolates (lineage A mpox2018 and lineage B.1 mpox2022) were titrated on LFDs in half-log serial dilutions in buffer 1 to give concentrations ranging from an initial $10^6$ to $10^{3.5}$ pfu per sample. As previously, samples were applied to LFDs in 100 µL volumes, and MVA virus was used as a positive control (Fig. 5). Consistent with previous experiments, the control band (C) is constant for all samples and the presence of viral antigen is detected at the test position from $10^6$ to $10^{4.5}$ pfu per sample. For reference purposes, we have included the MVA LOD (from Fig. 2) to show that the read-by-eye LOD for MPXV was identical to that seen with MVA ($3 \times 10^4$ pfu per sample).

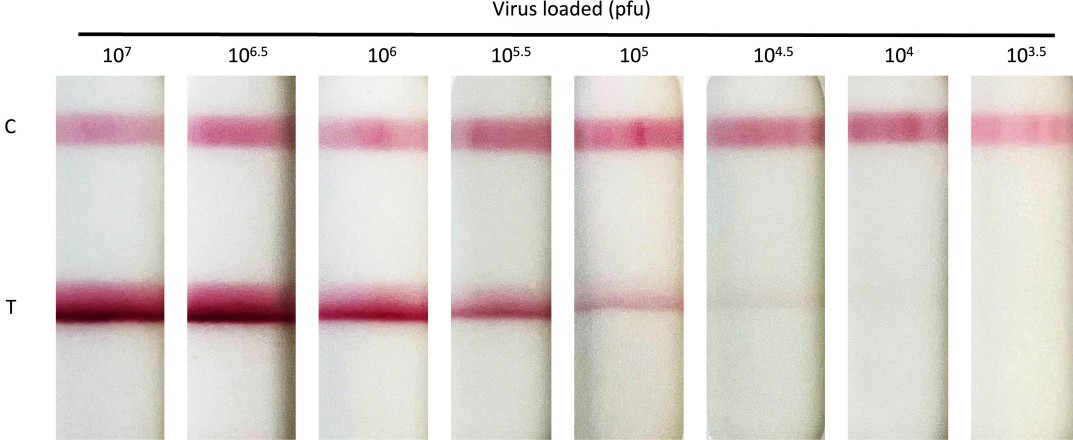

**FIG 2** Detection of vaccinia virus (MVA) using LFD. Vaccinia virus (MVA) was serially diluted to achieve virus concentrations ranging from $1 \times 10^7$ to $1 \times 10^{4.5}$ pfu/mL. Diluted virus (100 µL) was then applied to the LFD and development recorded at 20 minutes. The C band indicates that the LFD test has worked, and the T band indicates virus detection. Dilutions were carried out in CSB.

**TABLE 1** Composition of sample buffers supplied by the manufacturer

| Buffer ID | Components |
|---|---|
| 1 | 20 mM Tris, 100 mM NaCl, 1 mM EDTA, 0.5% Triton X-100 |
| 2 | 10 mM HEPES, 150 mM NaCl, 0.01% Tween-80, 0.1% sodium azide, 0.02% SDS |
| 3 | 10 mM HEPES, 150 mM NaCl, 0.01% Tween-80, 0.1% sodium azide, 0.1% SDS |
| 4 | PBS, 0.02% SDS |
| 5 | PBS, 0.1% SDS |
| 6 | TBS[a], 1% BSA[b], 0.1% Tween-80, 1.5% NaCl, 2 mM EDTA |
| 7 | 10 mM HEPES, 150 mM NaCl, 0.01% Tween-80, 0.01% sodium azide |

[a]Tris-buffered saline.
[b]Bovine serum albumin.

In order to determine the utility of the LFD as a POC, we used a set of stored frozen clinical skin swab samples (sent in viral transport medium), received for screening purposes during the 2022 MPXV outbreak. These samples had previously been tested by qRT-PCR to determine patient infection status and were all confirmed MPXV clade IIb. We selected 80 clinical samples (40 low Ct [between 10 and 20], 20 medium Ct [between 20 and 30], 20 high Ct [between 30 and 40]). In addition, to provide a differential diagnostic, 20 samples negative for MPXV, but positive for varicella virus or human papillomavirus, were included.

The genome copy number was evaluated in these samples using a bespoke RT-PCR assay based on the MPXV gene O2L. As shown in Fig. 6A, a linear relationship between Ct and genome copy number was established. Unfortunately, we were unable to attempt virus isolation and correlate Ct values to infectious units as all the samples had been inactivated following collection in inactivating viral transport medium (VTM).

In order to test these samples using the LFD, the 80 clinical and 20 negative samples were buffered by the addition of 10× LFD assay buffer 1 (Table 1), and 200 µL sample was added to the LFD. We were unable to detect any positive band at the test position, even in the sample with a Ct value of 14 (approximately $1.6 \times 10^8$ genomes per milliliter). In order to determine if there was an inhibitory chemical within the sample, the Ct14 sample was diluted 1:100 with either PBS or 1× LFD assay buffer 1 (Table 1, data not shown). The diluted sample gave a positive band at the test position on the LFD (see Fig. 6B). This suggests that sensitivity on the LFD for clinical samples is at least $3.2 \times 10^5$ when

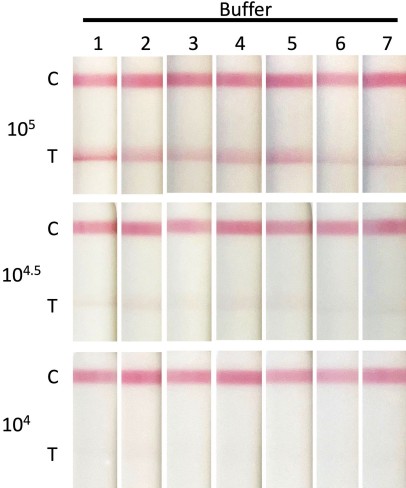

**FIG 3** Comparison of sample buffer to detect vaccinia virus (MVA) using LFD. Vaccinia virus (MVA) was serially diluted to achieve virus concentrations ranging from $1 \times 10^6$ to $1 \times 10^5$ pfu/mL. Diluted virus (100 µL) was then applied to the LFD and development recorded at 20 minutes. The C band indicates that the LFD test has worked, and the T band indicates virus detection. Dilutions were carried out in manufacturer-supplied sample buffers 1–7 (Table 1).

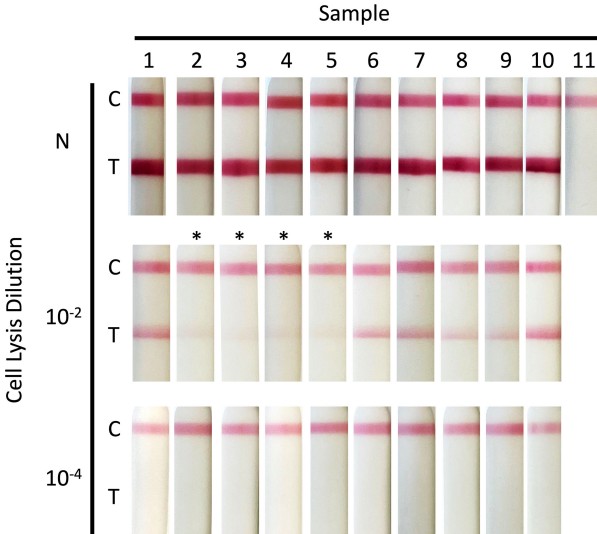

**FIG 4** Detection of cell-associated vaccinia virus (MVA) using LFD. Chicken fibroblast DF1 cells were infected with vaccinia virus (MVA) (MOI 1). Twenty-four hours post-infection, media was removed and cells were washed with PBS before lysis was performed using 1 mL of buffer (Table 2) followed by 5 minutes of incubation. Buffer and remaining cells were removed using a cell scraper. Sample 9 was also subjected to freeze thaw at −20°C. Neat (N) and diluted ($10^{-2}$, $10^{-4}$) cell lysate (100 µL) was then applied to the LFD and development recorded at 20 minutes. The samples containing SDS are indicated by *. The C band indicates that the LFD test has worked, and the T band indicates virus detection.

diluted 1:100. This value is 10-fold higher than purified virus ($3 \times 10^4$) and may reflect either the irreversible impact of the inhibitory component within the VTM on the viral epitope recognized by the mAbs and/or the differences between genome equivalent per milliliter and pfu per milliliter.

Guanidinium thiocyanate (GITC) is a common denaturing ingredient in inactivating VTM (42). Due to the reversible inhibitory effect of inactivating VTM on the detection of antigen when the clinical samples were applied to the LFD, we investigated the addition of GITC to LFD sample buffer 1. The presence of ≥1M GITC in the sample buffer masked the binding between antigen and antibody (see Fig. 7A). For samples containing 5M GITC, the effect was so great that the control band was not visible. We had previously seen that when we diluted the Ct14 clinical sample 1:100 (see Fig. 6) with PBS, a positive band was seen at the test position on the LFD. To test if we could reverse the masking by GITC, samples that had been prepared with 1M GITC were diluted by the addition of PBS and applied to the LFD. As can be seen in Fig. 7A, dilution was able to reverse the

**TABLE 2** Composition of lysis buffers

| Sample ID | Virus | Lysis components |
|---|---|---|
| 1 | +[a] | 20 mM Tris, 100 mM NaCl, 1 mM EDTA, 0.5% Triton X-100 |
| 2 | + | 10 mM HEPES, 150 mM NaCl, 0.01% Tween-80, 0.1% sodium azide, 0.02% SDS |
| 3 | + | 10 mM HEPES, 150 mM NaCl, 0.01% Tween-80, 0.1% sodium azide, 0.1% SDS |
| 4 | + | PBS, 0.02% SDS |
| 5 | + | PBS, 0.1% SDS |
| 6 | + | TBS, 1% BSA, 0.1% Tween-80, 1.5% NaCl, 2 mM EDTA |
| 7 | + | 10 mM HEPES, 150 mM NaCl, 0.01% Tween-80, 0.01% sodium azide |
| 8 | + | PBS |
| 9 | + | PBS + freeze thaw |
| 10 | + | Covid buffer (CSB) |
| 11 | −[b] | PBS |

[a]+, presence of virus
[b]−, absence of virus

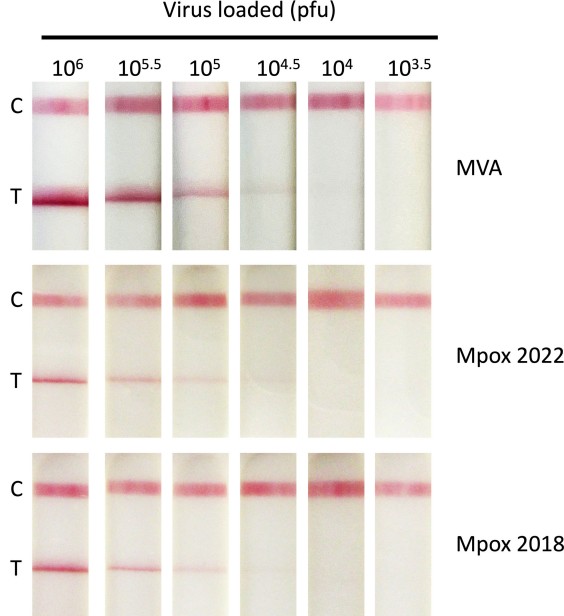

**FIG 5** Detection of mpox using LFD. Purified mpox virus (MPXV) strains isolated in 2018 and 2022 were serially diluted to achieve virus concentrations ranging from $1 \times 10^7$ to $1 \times 10^{4.5}$ pfu/mL. Diluted virus (100 µL) was then applied to the LFD and development recorded at 20 minutes. The C band indicates that the LFD test has worked, and the T band indicates virus detection. Dilutions were carried out in clinical sample buffer 1. For comparison, MVA (from Fig. 2) is included in panel 1.

masking effect of the GITC. As a number of viral transport media contain fetal bovine serum (FBS), we also tested the effect of using VTM as the assay buffer as well as adding FBS to LFD sample buffer 1. As can be seen in Fig. 7B, the sensitivity was $1 \times 10^6$ pfu/mL when the virus was diluted in VTM, which is approximately 0.5 log lower than using sample buffer 1. Addition of FBS to sample buffer 1 had no effect on the sensitivity of detection.

## DISCUSSION

The monoclonal antibodies in this LFD were originally developed for research use. We have assessed their repurposed potential for POC use in detecting MPXV in clinical samples to provide a diagnostic resource for disease control in outbreak situations. The majority of viral particles present at a lesion site are expected to be IMV. The presence of A27, an OPXV conserved protein, on the surface of IMV is therefore the most appropriate target antigen for an MPXV POC LFD. The highly conserved nature of A27 means that this LFD would be suitable for the detection of emerging strains of both clades I and II MPXV, including the novel clade Ib that has emerged in the DRC.

The sensitivity of our LFD has been assessed at $3 \times 10^4$ pfu ($3 \times 10^5$ pfu/mL); it is clear that this is sufficient for detection of virus in lesion swabs, which are the most likely sample types for POC diagnostic approaches. An earlier study with a commercial LFD (Tetracore Orthopox BioThreat Alert) detected virus antigen in 3/4 mpox lesion swabs with a virus titer as low as $1 \times 10^1$ pfu/mL, and 2/3 VACV lesion swabs with a virus titer as low as $1.2 \times 10^5$ pfu/mL. The LOD for the Tetracore LFD against purified VACV was $\sim 1.5 \times 10^6$ pfu per sample. This suggests that the particle per pfu ratio in swab material is at least one order of magnitude higher than for purified virus, or the swab material has a large quantity of free antigen, or both. In addition to the the lower LOD for our LFD vs the Tetracore LFD for purified virus, the mpox swab material tested on the Tetracore LFD was prepared with a protocol that included sonication (32), which is not optimal for in-home or POC use, suggesting that our LFD may be a stronger candidate for a POC diagnostic.

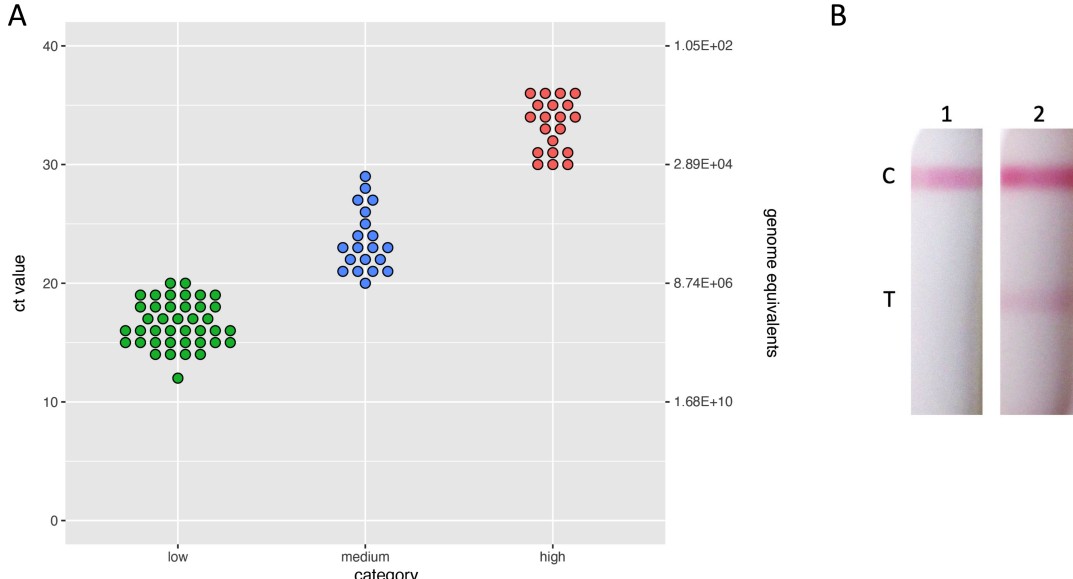

**FIG 6** Analysis of clinical samples. (A) One hundred clinical swab samples were stratified into negative (chickenpox or HPV positive), low (Ct between 10 and 20), medium (Ct between 20 and 30), and high (Ct between 30 and 40) Ct and subjected to MPXV-DNA quantification by real-time PCR. (B) MPXV clinical sample Ct14 applied undiluted (1) or diluted 1:100 with PBS (2).

One of the limitations of the LFD as an mpox diagnostic is that it detects all old-world OPXVs. An LFD specific for mpox would require mAbs targeting MPXV-specific epitopes. Although Dubois et al. (43) have used MPXV-specific peptides from the B21 protein to detect antibodies in human sera in ELISAs, there are no data to show that B21 is accessible on the viral particle. OPXV structural proteins are highly conserved so it may not be possible to design an LFD to distinguish between OPXVs when detecting virus particles in swabs. Although an LFD to detect antibodies against mpox in patient serum could have potential as a diagnostic, its utility in-home or as a POC would be hampered by a requirement for blood samples with a greater logistic burden than lesion swabs, and would be dependent on the kinetics of the antibody response, which is delayed by more than 8 days relative to rash eruption (44). As IgG has a long-lived presence in human sera, this approach may be better suited for a retrospective epidemiological survey.

A number of studies have compared mpox viral load (genomes) and infectious virus in clinical specimens (38, 45). The highest viral load is in lesion samples, two orders of magnitude higher than any other sample type (46). In studies quantifying infectious virus, virus was isolated from the majority of lesion samples, with approximately 70% of samples with a Ct ~26, equating to $3 \times 10^6$ genomes per milliliter, yielding viable virus. Assuming a particle per pfu ratio of 100:1, this would suggest our LOD of $3 \times 10^4$ pfu per sample, where a sample is 100 µL volume and is an order of magnitude lower than is required for detection of virus in lesion swabs. However, this calculation does not account for the possibility of unencapsidated virus DNA, which would overestimate the number of particles relative to Ct, or the possibility that a significant quantity of LFD-detectable non-viable particles or free antigen is present, which would make the LFD more sensitive relative to infectious virus. Paran et al. reported that a lesion Ct <22 gave a pfu per milliliter viral titer >$3 \times 10^5$ pfu/mL which suggests that those samples containing high numbers of viral particles would be detected by LFD (38). In agreement with this published data, we have shown that for samples with <Ct20 these contain $8.74 \times 10^6$/mL genome equivalents (Fig. 6).

Extrapolating LOD for purified virus to potential sensitivity against clinical material, by reconciling comparisons of Ct score against virus titer across multiple studies with different clinical material and preparation/extraction protocols, is challenging. An inherent problem with using the available clinical samples which were initially collected

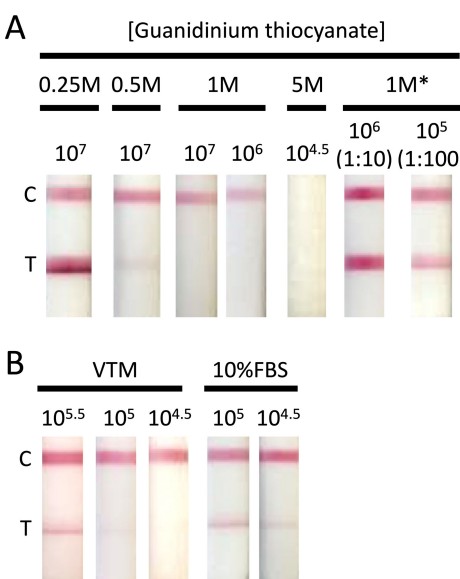

**FIG 7** Effect of guanidinium thiocyanate on antibody:antigen interaction. (A) Vaccinia virus (MVA) was serially diluted to achieve virus concentrations ranging from $1 \times 10^8$ to $1 \times 10^{5.5}$ pfu/mL. Dilutions were carried out in (A) manufacturer-supplied sample buffer 1 (Table 1) with the addition of guanidinium thiocyanate at a concentration between 5M and 0.25M. Samples marked with * were prepared in 1M guanidinium thiocyanate at $10^8$ pfu/mL, incubated at room temperature for ~1 hour, and then diluted either 1:10 or 1:100 with PBS. (B) VTM (BD Biosciences) or manufacturer-supplied sample buffer 1 (Table 1) with the addition of 10% fetal bovine serum. Diluted virus (100 µL) was applied to the LFD and development recorded at 20 minutes. The C band indicates that the LFD test has worked, and the T band indicates virus detection.

for MPXV qPCR screening purposes is that the buffers are optimized for that assay and are not suitable for testing with the LFD. Due to the infectivity of MPXV, for safety purposes, samples are usually collected in inactivating VTM. These inactivating VTMs commonly contain guanidinium thiocyanate (42). In the case of the frequently used Copan eNat, the concentration of GITC is 3.67M. Although these buffers are ideal for use in situations where nucleic acids need to be preserved for PCR, they are also known to denature proteins (47), disrupting Ag and Ab interaction. We have shown that inactivating VTM contains an inhibitory component that can cause antigen masking. This results in a false negative with no band detectable. For the highest viral load sample in our study (Ct14), we were unable to detect Ag when the clinical sample was in transport media, but this was resolved when we diluted the sample 1:100 with either PBS or LFD dilution buffer. The addition of denaturant to the LFD sample buffer used in this study resulted in antigen masking at concentrations above 0.5M GITC. This suggests that a study of different VTMs is needed to determine the optimal sampling reagent and thus LFD performance. Other inactivating medium, formulated using, e.g., ammonium sulfate (48), may be better suited to LFD use. The addition of fetal bovine serum, commonly found in non-inactivating VTM, had little effect on Ag detection.

To be certain of the utility of our LFD, it will require testing against a bank of clinical samples taken in an LFD-optimized buffer, with robust metadata on age of lesion. As we expect this LFD to be used at POC, it should not be necessary to use inactivating VTM. We are currently working to secure access to a suitable archive of material for testing. Ideally, this archive would include bodily fluids such as urine and saliva to demonstrate the robustness of the LFD.

Extrapolating from previously published data (49) and those presented here, we predict that the current sensitivity will detect MPXV in swab samples shown to be <Ct22. This could be improved by using antibodies against both IMV and EV antigens (F13L is the most abundant protein in the VACV EV envelope [50]). Clarification of the

ratio of IMV to EV in MPXV infections would determine if this would be a sensible approach. Additionally, technology has improved since the design of the LFD used in this study (51). Advanced nanoparticles that allow signal enhancement could be utilized to increase the strength of the testing line. Novel approaches to tagging antigens with DNA sequences and enriching the samples also hold potential (51). This has the advantage of potentially providing a differentiation between MPXV and other OPVs. Alternative storage and running buffers that preserve the MPXV antigen and ensure minimal degradation of the antigen may be essential for any LFD to test for MPXV given poor clinical performances to date. Any innovation in LFD design would need to be compatible with the aim of providing REASSURED and affordable diagnostics in remote areas (52). The data presented here were interpreted by reading the LFD by eye. We recognize that interpretation of the results may vary from person to person depending on many factors, and may lead to misreading. This could be overcome by the use of technology such as smartphone apps (53, 54). As long as these aids were free and available to all users, we would welcome this development.

There is clear potential for MPXV (clades I and II), circulating in humans only, in particular, demographics, to adapt to more general transmission, as occurred with other OPXVs such as variola virus in humans and CMLV in camels. If this should occur for mpox, then a robust POC diagnostic will be a major component of control measures, and the absence of such a diagnostic would be keenly felt. It is thus important that the utility of LFDs as POC diagnostics for mpox is tested in a timely fashion, particularly given the recent emergence and rapid spread of clade I mpox in DRC and neighboring areas.

## ACKNOWLEDGMENTS

This work was supported by US Food and Drug Administration Medical Countermeasures Initiative contract 75F40120C00085 and the Pandemic Sciences Institute (using funding provided by AstraZeneca UK Ltd.). Work in C.M.D.M. laboratory was funded by UK Biotechnology and Biological Sciences Research Council grant BB/X0011356/1.

## AUTHOR AFFILIATIONS

[1]Pandemic Sciences Institute (PSI), University of Oxford, Oxford, United Kingdom
[2]Centre for Human Genetics (CHG), University of Oxford, Oxford, United Kingdom
[3]DSTL, Salisbury, United Kingdom
[4]Department of Microbial Sciences, University of Surrey, Guildford, United Kingdom
[5]Department of Tropical Disease Biology, Liverpool School of Tropical Medicine, Liverpool, United Kingdom
[6]Global Access Diagnostics, Bedford, United Kingdom
[7]Laboratory of Clinical and Epidemiological Virology (Rega Institute), Ku Leuven, Leuven, Belgium

## AUTHOR ORCIDs

Stephen M. Laidlaw http://orcid.org/0000-0002-7970-6174
Rebecca Sumner http://orcid.org/0000-0003-0735-8649
Thomas Edwards http://orcid.org/0000-0003-4058-4461
Carlos Maluquer de Motes http://orcid.org/0000-0003-4712-4601
Miles W. Carroll http://orcid.org/0000-0002-7026-7187

## FUNDING

| Funder | Grant(s) | Author(s) |
| --- | --- | --- |
| MOHW \| Food and Drug Administration (FDA) | 75F40120C00085 | Miles W. Carroll |

| Funder | Grant(s) | Author(s) |
| --- | --- | --- |
| UKRI \| Biotechnology and Biological Sciences Research Council (BBSRC) | BB/X0011356/1 | Carlos Maluquer de Motes |

## AUTHOR CONTRIBUTIONS

Stephen M. Laidlaw, Investigation, Methodology, Writing – original draft | David Ulaeto, Conceptualization, Data curation, Methodology, Writing – review and editing | Steve Lonsdale, Methodology, Writing – review and editing | Graeme Clark, Project administration, Writing – review and editing | Rebecca Sumner, Formal analysis, Methodology, Writing – review and editing | Thomas Edwards, Methodology, Writing – review and editing | Emily Adams, Formal analysis, Writing – review and editing | Anne-Sophie Logist, Investigation, Writing – review and editing | Bram Van Holm, Investigation, Writing – review and editing | Carlos Maluquer de Motes, Formal analysis, Investigation, Writing – review and editing, Funding acquisition | Peter Horby, Funding acquisition, Resources, Writing – review and editing | Piet Maes, Conceptualization, Investigation, Methodology, Writing – review and editing | Miles W. Carroll, Conceptualization, Funding acquisition, Writing – review and editing

## ADDITIONAL FILES

The following material is available online.

Open Peer Review

**PEER REVIEW HISTORY (review-history.pdf).** An accounting of the reviewer comments and feedback.

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
