## [Reviewer comments · Microbiology Spectrum]

Microbiology Spectrum

Detection of Mpox and other Orthopoxviruses using a Lateral Flow Device as a Point of Care diagnostic

Stephen Laidlaw, David Ulaeto, Steve Lonsdale, Graeme Clark, Rebecca Sumner, Thomas Edwards, Emily Adams, Anne-Sophie Logist, Bram Van Holm, Carlos Maluquer de Motes, Peter Horby, Piet Maes, and Miles Carroll

Corresponding Author(s): Miles Carroll, University of Oxford

Review Timeline:

Submission Date:	September 27, 2024
Editorial Decision:	November 23, 2024
Revision Received:	December 20, 2024
Accepted:	December 23, 2024

Editor: Benjamin Liu

Reviewer(s): The reviewers have opted to remain anonymous.

Transaction Report:

DOI: <https://doi.org/10.1128/spectrum.02456-24>

Re: Spectrum02456-24 (Detection of Mpox and other Orthopoxviruses using a Lateral Flow Device as a Point of Care diagnostic.)

Dear Prof. Miles W Carroll:

Thank you for the privilege of reviewing your work. Below you will find my comments, instructions from the Spectrum editorial office, and the reviewer comments.

Editor's comments:

1. Introduction: "However, this method (PCR) is not suitable for home or lower-resource POC diagnosis and the resulting delays can hamper efforts to slow or halt disease spread.": In this paragraph, there are no any references to support the authors' statement. The authors are recommended to cite published papers on the drawbacks of PCR for mpox, eg, lowered sensitivity and misidentification due to atypical mpox presentations. More references should be cited, with this one (PMID: 38793665) as an example (citing is optional).
2. What is the definition of LOD in this study? LOD determination needs triplicate testing for the same concentration. Did the authors do this?
3. For faint bands shown in figures, there is a chance for human error of misreading.
4. Abstract should be added with more results.

Please return the manuscript within 30 days; if you cannot complete the modification within this time period, please contact me. If you do not wish to modify the manuscript and prefer to submit it to another journal, notify me immediately so that the manuscript may be formally withdrawn from consideration by Spectrum.

Revision Guidelines

Sincerely,
Benjamin Liu
Editor
Microbiology Spectrum

Reviewer #1 (Comments for the Author):

In 2022, the global outbreak of Mpox poses a serious threat to human health, and due to its wide spread, it will be a persistent threat in the coming years. Developing rapid and accurate diagnostic reagents is crucial for preventing and diagnosing Mpox and other Orthopoxviruses. Especially in the early stages of infection, the superiority of antigen detection is demonstrated.

The paper conducted a study on the antigen detection methodology of A27 protein, and conducted detailed detection and analysis from different perspectives and levels such as antibody characteristics, detection limits, different virus strains, dilution buffer solutions, clinical samples, especially regarding the impact of nucleic acid detection reagents on antigen detection. This laid a foundation for the subsequent study of virus pathogenesis, vaccine research, and preventive control.

The paper has rigorous logic and clear arguments, making it a meaningful research with value of real-world application.

Reviewer #2 (Comments for the Author):

Figure 1A: Titration of Two mAbs

While the symbols for each line are provided in the legend below the figure, I suggest adding labels directly to the lines on the graph for improved clarity. The linear range appears to be approximately 5×10^4 to 8×10^6 PFU/well, which is later confirmed in Figure 2.

Stability Testing

To assess stability, the authors tested different buffers. Notably, a concentration of $10^{4.5}$ PFU/well resulted in visible bands, while 10^5 PFU/well produced clear bands. Based on these observations, the lower detection limit for this method seems to be $10^{4.5}$ PFU/well. This threshold is relatively high in my opinion. While this limitation is discussed in the manuscript, I would recommend including a comparison with other available testing methods to provide additional context and justification.

Application of the Method

The authors applied the method to various sample types, including cell-associated MVA and MPXV, as well as frozen clinical skin swab samples. The method yielded reasonably consistent results across these different sample types.

Response to Reviewers: "Detection of Mpox and other Orthopoxviruses using a Lateral Flow Device as a Point of Care diagnostic" (Spectrum02456-24).

Editor's comments:

1. Introduction: "However, this method (PCR) is not suitable for home or lower-resource POC diagnosis and the resulting delays can hamper efforts to slow or halt disease spread.": In this paragraph, there are no any references to support the authors' statement. The authors are recommended to cite published papers on the drawbacks of PCR for mpox, eg, lowered sensitivity and misidentification due to atypical mpox presentations. More references should be cited, with this one (PMID: 38793665) as an example (citing is optional).

Response: Thank you for your comments. In order to address this point, we have added the following:

Current CDC guidelines recommend that testing for mpox is only carried out for people with a rash consistent with mpox [24, 25]. In the current outbreak, mpox lesions are reported to often be atypical [26, 27]. This may be due to un-diagnosed cases of co-infection [24, 28, 29]. The presence of a co-incident rash that may present at the same location as the mpox rash can result in mis-diagnosis or under-reporting following PCR, due to either lower sensitivity or mis-identification.

2. What is the definition of LOD in this study? LOD determination needs triplicate testing for the same concentration. Did the authors do this?

Response: Thank you for bringing this to our attention. We have added the following sentences:

We defined this LOD as the lowest amount of virus that resulted in a visible band when viewed by eye. Confirmation of the LOD was carried out in all subsequent experiments (n=4) as an internal control (data not shown).

3. For faint bands shown in figures, there is a chance for human error of misreading.

Response: We have added the following:

The data presented here was interpreted by reading the LFD by eye. We recognise that interpretation of the results may vary from person to person depending on many factors, and may lead to misreading. This could be overcome by the use of technology such as smartphone apps [53, 54]. As long as these aids were free and available to all users, we would welcome this development.

4. Abstract should be added with more results.

Response: We have added the following paragraph:

We found that the presence of guanidinium thiocyanate, a common ingredient in inactivating viral transport media, masked the LFD antigen resulting in false negatives. We therefore conclude that an LFD optimised buffer is required, as the sample collection buffer may have a detrimental impact on sensitivity for clinical material.

Reviewer #2 (Comments for the Author):

Figure 1A: Titration of Two mAbs

While the symbols for each line are provided in the legend below the figure, I suggest adding labels directly to the lines on the graph for improved clarity. The linear range appears to be approximately 5×10^4 to 8×10^6 PFU/well, which is later confirmed in Figure 2.

Response: Thank you for your suggestion. We tried to add labels to the lines but unfortunately the result was not satisfactory. Instead, we have included a legend directly below Fig A.

Stability Testing

To assess stability, the authors tested different buffers. Notably, a concentration of $10^{4.5}$ PFU/well resulted in visible bands, while 10^5 PFU/well produced clear bands. Based on these observations, the lower detection limit for this method seems to be $10^{4.5}$ PFU/well. This threshold is relatively high in my opinion. While this limitation is discussed in the manuscript, I would recommend including a comparison with other available testing methods to provide additional context and justification.

Response: Unfortunately, it has proven universally difficult to produce an LFD suitable for high sensitivity mpox detection. The LFD that we have developed is currently the most sensitive of all current mpox LFDs. We are currently working to increase this sensitivity using new methods of coupling etc. The requirement for an inexpensive home POC precludes technologies such as PCR and Luminex and as such a comparison with methods such as these would not be practical.

Re: Spectrum02456-24R1 (Detection of Mpox and other Orthopoxviruses using a Lateral Flow Device as a Point of Care diagnostic.)

Dear Prof. Miles W Carroll:

Your manuscript has been accepted, and I am forwarding it to the ASM production staff for publication. Your paper will first be checked to make sure all elements meet the technical requirements. ASM staff will contact you if anything needs to be revised before copyediting and production can begin. Otherwise, you will be notified when your proofs are ready to be viewed.

Sincerely,
Benjamin Liu
Editor
Microbiology Spectrum